# NMGrad: Advancing Histopathological Bladder Cancer Grading with Weakly Supervised Deep Learning

**DOI:** 10.3390/bioengineering11090909

**Published:** 2024-09-11

**Authors:** Saul Fuster, Umay Kiraz, Trygve Eftestøl, Emiel A. M. Janssen, Kjersti Engan

**Affiliations:** 1Department of Electrical Engineering and Computer Science, University of Stavanger, 4021 Stavanger, Norway; trygve.eftestol@uis.no (T.E.); kjersti.engan@uis.no (K.E.); 2Department of Pathology, Stavanger University Hospital, 4011 Stavanger, Norway; umay.kiraz@sus.no (U.K.); emilius.adrianus.maria.janssen@sus.no (E.A.M.J.); 3Department of Chemistry, University of Stavanger, 4021 Stavanger, Norway

**Keywords:** computational pathology, deep learning, grading, multiscale, urothelial carcinoma, weakly supervised learning

## Abstract

The most prevalent form of bladder cancer is urothelial carcinoma, characterized by a high recurrence rate and substantial lifetime treatment costs for patients. Grading is a prime factor for patient risk stratification, although it suffers from inconsistencies and variations among pathologists. Moreover, absence of annotations in medical imaging renders it difficult to train deep learning models. To address these challenges, we introduce a pipeline designed for bladder cancer grading using histological slides. First, it extracts urothelium tissue tiles at different magnification levels, employing a convolutional neural network for processing for feature extraction. Then, it engages in the slide-level prediction process. It employs a nested multiple-instance learning approach with attention to predict the grade. To distinguish different levels of malignancy within specific regions of the slide, we include the origins of the tiles in our analysis. The attention scores at region level are shown to correlate with verified high-grade regions, giving some explainability to the model. Clinical evaluations demonstrate that our model consistently outperforms previous state-of-the-art methods, achieving an F1 score of 0.85.

## 1. Introduction

Bladder cancer, a prevalent urological malignancy, poses significant clinical challenges, in terms of both diagnosis and prognosis [1]. Non-muscle-invasive bladder cancer (NMIBC) accounts for approximately 75% of the newly diagnosed cases of urothelial carcinoma. NMIBC is particularly known for its variable outcomes, necessitating accurate and consistent grading for optimal patient management [2]. The 2022 edition of the European Association of Urology guidelines on NMIBC recommends a stratification of patients into risk groups based on the risk of progression to muscle-invasive disease [3]. Grade, stage, and various other factors contribute to the risk. Precise risk assessment is vital in the management of NMIBC, since treatment strategies do not only rely on the presence of muscle invasion.

Grading is based on assessment of the cellular morphology abnormalities of urothelial tissue. In 2004, the WHO introduced a grading classification system (WHO04) for NMIBC, based on histological features. WHO04 encompasses three categories: papillary urothelial neoplasm of low malignant potential (PUNLMP); non-invasive papillary carcinoma low-grade (LG); and non-invasive papillary carcinoma high-grade (HG), ranging from lower to higher malignancy, respectively [4]. HG is related to lower differentiation, loss of polarity, and pleomorphic nuclei, among others. The intricate evaluation of heterogeneous scenarios contributes to significant inter- and intra-observer variability. Disparities potentially lead to misclassification and, consequently, to inappropriate treatment decisions [5]. The WHO04 grading system was subsequently retained in the updated 2016/2022 WHO classifications [6]. However, according to several multi-institutional analyses of individual patient data, the proportion of tumors classified as PUNLMP (WHO 2004/2016) has markedly decreased to very low levels in the last decade. This trend has resulted in the suggestion to reassess PUNLMP tumors as LG [7,8]. Therefore, the upcoming grading system will reasonably undergo a modification, shifting towards the inclusion of only LG and HG categories. In recent years, the integration of deep learning techniques within the field of computational pathology (CPATH) has offered promising avenues for enhancing the precision of computer-aided diagnosis (CAD) systems and elucidating discrepancies among pathologists [9,10]. Consequently, the synergy between histological expertise and CAD technologies is vital for accurate grading assessments.

CPATH is the field of pathology that leverages the potential of CAD systems to thoroughly analyze high-resolution digital images known as whole slide images (WSIs) for diverse diagnostic and prognostic purposes [11]. WSIs are produced by slide scanners and pre-stored at different magnification levels, emulating the functionality of physical microscopes. Lower magnification is suitable for tissue-level morphology examination, while higher magnification is suitable for cell-level scrutiny [12]. WSIs are characterized by their substantial size, which can introduce adversarial noise. Bladder cancer WSIs present unique challenges, due to their disorganized nature and the presence of diagnostically non-relevant tissue. These slides often include artifacts, such as cauterized or stretched tissue [13]. Moreover, tissue such as blood, muscle, and stroma are less informative for grading a tumor. Therefore, the absence of annotations presents a significant challenge for identifying regions of interest (ROIs) [9]. It is crucial to distinguish between region-based labels (e.g., tissue type, grade) and WSI-based labels, including follow-up information and overall patient grade. Grade exemplifies a label that encompasses both perspectives [14,15]. While clinical reports assign the worst grade observed to a patient, a WSI may exhibit diverse urothelium regions with normal, LG, and HG. This dual nature underscores the complexity of label interpretation when considering both the medical, WSI-focused perspective and the more technical perspective involving regional data analysis and processing.

CPATH is in a transformative era, aiming to reshape the landscape of digital pathology as we know it [16]. Among the diverse practices in the field, imaging methodologies rooted in convolutional neural networks (CNNs) have emerged as the foundation of feature extraction from histological images [11]. These deep learning networks possess a remarkable capacity to automatically discern morphological and cellular patterns within WSIs [17,18,19,20,21,22]. Ultimately, CNNs contribute to more precise and timely clinical decisions. However, training deep learning models in CPATH presents challenges when only WSI-level labels are available, lacking region-based annotations [9,23,24,25,26]. To address these, weakly supervised learning techniques, like attention-based methods and multiple-instance learning (MIL), are employed. However, MIL methods can be susceptible to individual instances dominating the weighted aggregation of the WSI representation [27,28,29]. In the context of WSIs, the tissue is distributed across the slide, for which reason the regions typically present similar features and pathologists are able to pinpoint ROIs with crucial information [30]. Specifically, while grading, situations may arise where multiple instances in close proximity exhibit HG characteristics, while other regions may concurrently display LG attributes. As a result, constraining instances to specific regions enhances our understanding of the diverse features within WSIs. Consequently, a conventional MIL architecture approach may not be appropriate, because there is a susceptibility to information leakage between regions. A model accommodating the nested structure of WSIs—wherein tissues are part of a region, and regions, in turn, belong to a WSI—may more effectively capture the clinical WSI-level grading label [31,32,33].

In this study, we introduce a novel pipeline for grading NMIBC using histological slides, referred to as nested multiple grading (NMGrad). The proposed solution starts by tissue segmentation of the WSI, separating urothelium from other tissue types. The next step categorizes extracted tiles of urothelium areas into location-dependent regions for predicting the patient’s WHO04 grade. We implemented a weakly supervised learning framework, using attention mechanisms and a nested aggregation architecture for ROI differentiation. Our method offers an innovative approach for generating diagnostic suggestions, with generated heatmaps for highlighting tiles and ROIs independently.

## 2. Related Work

Numerous studies in the domain of computational pathology for bladder cancer diagnostics have emerged in recent years [34]. In Wetteland et al. [35], a pipeline for grading NMIBC was introduced. This pipeline identifies relevant areas in the WSIs and predicts the cancer grade by considering individual tile predictions and applying a decision threshold to determine the overall patient prediction. Their results demonstrated promising performance, with potential benefits for patient care. In Zheng et al. [36], the authors focused on the development of deep learning-based models for bladder cancer diagnosis and predicting overall survival in muscle-invasive bladder cancer patients. They introduced two deep learning models for diagnosis and prognosis, respectively. They showed that their presented algorithm outperformed junior pathologists. In Jansen et al. [37], the authors proposed a fully automated detection-and-grading network based on deep learning, to enhance NMIBC grading reproducibility. The study employed a U-Net-based segmentation network to automatically detect urothelium, followed by a VGG16 CNN network for classification. Their findings demonstrated that the automated classification achieved moderate agreement with consensus and individual grading from a group of three senior uropathologists. Spyridonos et al. [38] investigated the effectiveness of support vector machines and probabilistic neural networks for urinary bladder tumor grading. The results indicated that both SVM and PNN models achieve a relatively high overall accuracy, with nuclear size and chromatin cluster patterns playing key roles in optimizing classification performance.

Zhang et al. [39] addressed a common limitation of interpretability in CAD methods. To tackle this, they introduced MDNet, a novel approach that established a direct multimodal mapping between medical images and diagnostic reports. This framework consists of an image model and a language model. Through experiments on pathology bladder cancer images and diagnostic reports, MDNet demonstrated superior performance compared to comparative baselines. Zhang et al. [40] proposed a method that leverages deep learning to automate the diagnostic reasoning process through interpretable predictions. Using a dataset of NMIBC WSIs, the study demonstrated that their method achieves diagnostic performance comparable to that of 17 uropathologists.

Two critical challenges we have identified include summarizing information from local image features into a WSI representation and the scarcity of annotated datasets. Effectively translating detailed local information to the WSI level is complex, particularly in tasks like grading NMIBC. Moreover, the limited availability of well-annotated datasets hinders the development and evaluation of robust models. To tackle these issues, weakly supervised methods have emerged as a standout tool in CPATH [9]. While weakly supervised methods are widespread, some studies still rely on annotations and supervised learning. However, there is a growing consensus for the future of CPATH to predominantly embrace weakly supervised approaches. This shift is being driven by the impracticality of obtaining detailed annotations for large datasets covering various cancers and tasks. Among the various weakly supervised methods, attention-based MIL (AbMIL), a popular instance-aggregation method, exploits attention mechanisms, in order to mitigate the uncertainty from individual instances and enhance interpretability [41,42]. AbMIL bridges the gap between limited supervision and the spatial details necessary for accurate analysis and explainability. An evolution of MIL model architectures relies on the arrangement of the data within bags, where instances are further subdivided into finer groups. This concept is referred to as nesting [31,32]. Nested architectures preserve a sense of localization or categorization by selectively processing data instances within individual subgroups. Subsequently, they aggregate summarized information from the subgroups into a final bag representation.

In our work, we aimed to bridge the gap between non-annotated datasets, weakly supervised methods, and the intrinsic categorization of WSI data. Therefore, we leveraged the nested MIL with the attention mechanisms (NMIA) model architecture that we proposed in Fuster et al. [33], for accurate and interpretable NMIBC grading. Finally, in order to overcome the lack of annotations for defining the tissue of interest, a tissue segmentation algorithm TRI-25× -100× -400× was proposed by our research group in [43]. More recent works from our research group on tissue segmentation have found adoption within the scientific literature [44,45]. The utilization of this segmentation algorithm offers the opportunity to extract tiles specifically from the urothelium, contributing to a refined and targeted extraction process.

## 3. Data Material

The dataset comprised a total of 300 digital whole-slide images (WSIs) derived from 300 patients diagnosed with NMIBC, from the Department of Pathology, Stavanger University Hospital (SUH) [14,46]. The glass slides were digitized using a Leica SCN400 slide scanner and saved in the vendor-specific SCN file format. Collected over the period spanning 2002 to 2011, this dataset encompassed all risk group cases of non-muscle-invasive bladder cancers. The biopsies were processed through formalin fixation and paraffin embedding, and, subsequently, 4 µm thick sections were prepared and stained using hematoxylin, eosin, and saffron (HES). Furthermore, all WSIs underwent meticulous manual quality checks, ensuring the inclusion of only high-quality slides with minimal or no blur. Due to the cauterization process used in the removal of NMIBC, some slides exhibited areas with burned and damaged tissues. All WSIs originated from the same laboratory, resulting in relatively consistent staining color across the dataset.

All WSIs were graded by an expert uropathologist, in accordance with the WHO04 classification system, as either LG or HG, thus providing slide-level diagnostic information. However, the dataset lacked region-based annotations pinpointing the precise areas of LG or HG regions within the WSI. Consequently, the dataset was considered weakly labeled. For WSIs labeled as LG, at least one LG region was expected, with the possibility of presenting non-cancerous tissue in other regions. As for HG slides, at least one region should display HG tissue, while other regions may exhibit an LG appearance or non-cancerous tissue. Given the absence of alternative gold standards, we were compelled to continue utilizing a grading assessment that might have limitations for training and evaluating our algorithms. The dataset employed in this study was divided into three subsets: 220 WSI/patients for training, 30 for validation, and 50 for testing. The split employed ensured that each subset maintained the same proportional representation of diagnostic outcomes. This stratification encompassed factors such as WHO04 grading, cancer stage, recurrence, and disease progression, to best mirror the diversity of the original data material. The distribution of LG and HG WSIs within each dataset is detailed in Table 1, for reference.

Within a subset of the test set, denoted as Test_ANNO_ ∈ Test, 14 WSIs contained either one or two annotated regions of confirmed LG or HG tissue, verified by an expert uropathologist. It is noteworthy that not all regions were annotated. The labels of these regions corresponded to the associated weak label of the WSI.

## 4. Methods

We propose NMGrad, a pipeline that begins with a tissue segmentation algorithm for extracting urothelium tissue. Subsequently, the urothelium is divided into localized regions. Thereafter, we employ a weakly supervised learning method to predict tumor grade from the segmented urothelium regions. We exploit the sense of region locations by adopting a nested architecture with attention, NMIA [33]. The rationale behind employing this structured data arrangement analysis is to identify relevant instances and regions within the WSI. The attention mechanism and the nested bags/regions also contribute to a more precise and insightful analysis of the data. An overview of NMGrad is visualized in Figure 1:

### 4.1. Automatic Tissue Segmentation and Region Definition

We utilize the tissue segmentation algorithm introduced by Wetteland et al. [43] to automatically generate tissue type masks, facilitating the subsequent extraction of tiles. We define triplets T of tissue, which consist of a set of three tiles at various magnification levels, namely 25×, 100×, and 400×. An example is shown in Figure 2. We use a tile size of 128×128 for all magnification levels. Triplets are formed to maintain consistency, ensuring that the center pixel in every tile accurately represents the same physical point. The tissue segmentation algorithm works at tile level and classifies all triplets T in the WSI as y∈Y= {*urothelium, lamina propria, muscle, blood, damage, background*}. As grading relies on *urothelium* alone, we utilize the *urothelium* mask for defining valid areas for tile extraction, as described in [47]. In this work, various magnifications are explored for defining the model’s input, either using mono-scale MONO (400×), di-scale DI (400×, 100×) or tri-scale TRI (400×, 100×, 25×). We employ 400× magnification to establish a tight grid of tiles for data extraction purposes. These sets of tiles are later fed to the grading models, where each magnification tile is processed by its respective weight-independent CNN. To preserve the sense of location within the image, we define regions. This results in the following stratified division of data: all urothelium in the WSI, scattered regions of urothelium, and finally, individual tiles of urothelium.

#### Region Definition

For defining regions out of the extracted urothelium tiles, we define blobs of tiles ${\mathrm{URO}}_{\mathrm{BLOB}}\subseteq \mathrm{URO}$. ${\mathrm{URO}}_{\mathrm{BLOB}}$ is formed when tiles are 8-connected, and this joint set of tiles is the representation of a region ${\mathrm{URO}}_{\mathrm{BLOB}}=\{T_{1},T_{2}\ldots \}$. A region is eligible for inclusion if the number of tiles $N_{B}$ is higher than the threshold number $T_{\mathrm{LOWER}}$. Any blob with $N_{B}<T_{\mathrm{LOWER}}$ tiles is discarded, along with the tiles within. As NMIBC WSI can contain large tissue bundles, resulting in sizable blobs, we also define an upper-limit threshold $T_{\mathrm{UPPER}}$. For blobs where $N_{B}\geq T_{\mathrm{UPPER}}$, we split the region into several sub-regions for more detailed analysis. We apply KMeans clustering over the coordinates of the tiles x,y within the blob for location sense, defining the number of clusters as $N_{C}=\left\lceil N_{B}/T_{\mathrm{UPPER}}\right\rceil$. This results in joint regions within a bundle of tissue of consistent size, as observed in regions 5–8 in Figure 3.

### 4.2. Multiple-Instance Learning in a WSI Context

Multiple instance learning (MIL) is a weakly supervised learning method where unlabeled instances are grouped into bags with known labels [27,28]. A dataset $X,Y=\left\{(X^{i},y^{i}),\forall i=1,\ldots ,N\right\}$ is formed of pairs of sample sets X and their corresponding labels *y*, where *i* denotes a bag index. In the context of WSI, the bag can be one patient or one WSI or one region. In a conventional MIL data arrangement, we consider the bag X to be one WSI consisting of instances $x_{l}$:(1)$$X=\left\{x_{l},\forall l=1,\ldots ,L\right\}$$
where *L* is the number of instances in the bag. In our study, x refers to individual tiles in a set of extracted tiles from a patient slide X. A feature extractor $G_{\theta }:X\to H$ transforms image tiles, $x_{l}$, into low-dimensional feature embeddings, $h_{l}$. At this point, the bag structure previously formed remains intact, as instances have been simply transformed. Given a label *y* for a WSI, the training objective of the model is to predict the grade observed in the WSI. However, to deduce the specific region(s) within a WSI that leads to the patient’s diagnosis of either LG or HG is of the utmost importance. This entails the model’s ability to discern and highlight the critical areas within the WSI that play a pivotal role in the diagnosis. In order to accomplish this goal, we adopted attention-based multiple-instance learning (AbMIL) as our MIL framework, using attention-based aggregation, as shown in Figure 1. An attention score $a_{i}$ for a feature embedding $h_{i}$ can be calculated as
(2)$$a_{i}=\frac{\exp\{w^{⊤}\left(\tanh\left({\mathrm{Vh}}_{i}^{⊤}\right)⊙\mathrm{sigm}\left({\mathrm{Uh}}_{i}^{⊤}\right)\right)\}}{\sum_{l=1}^{L}\exp\left\{w^{⊤}\left(\tanh\left({\mathrm{Vh}}_{l}^{⊤}\right)⊙\mathrm{sigm}\left({\mathrm{Uh}}_{l}^{⊤}\right)\right)\right\}}$$
where $w\in R^{L\times 1}$, $V\in R^{L\times M}$, and $U\in R^{L\times M}$ are trainable parameters and ⊙ is an element-wise multiplication. Furthermore, the hyperbolic tangent tanh(·) and sigmoid sigm(·) are included, to introduce non-linearity for learning complex applications. The benefit of attention modules extends beyond interpretability for understanding the model’s decision-making process, as it also grants enhanced predictive capabilities by prioritizing salient features. This is because attention scores directly influence the forward propagation of the model, allowing it to focus on the most relevant and informative regions within the data. Once the attention scores A are obtained, we obtain the patient prediction $\hat{y}$, using a patient classifier $\Psi_{\rho }$, as
(3)$$\hat{y}=\Psi_{\rho }\left(H^{a}\right)=\Psi_{\rho }\left(A\cdot H\right)=\Psi_{\rho }\left(A\cdot G_{\theta }\left(X\right)\right)$$

#### Nested Multiple-Instance Architecture

An evolution of the conventional AbMIL architecture defines levels of bags within bags where only the innermost bags contain instances. This is referred to as nested multiple instance with attention (NMIA) [33]. A bag-of-bags for a WSI $H_{\mathrm{WSI}}$ contains a set of inner bags, or regions, $H_{\mathrm{REG},k}$:(4)$$H_{\mathrm{WSI}}=\left\{H_{\mathrm{REG},k},\forall k=1,\ldots ,K\right\}$$
where the number of inner bags *K* varies between different WSI. Ultimately, $H_{\mathrm{REG},k}$ contains instance-level representations $h_{\mathrm{TILE},l}$ of tiles located within the physical region. This serves to further stratify into clusters or regions and to accurately represent the arrangement of the scattered data, where tiles belong to particular tissue areas, and the areas themselves belong to the WSI:(5)$$h_{\mathrm{REG}}=A_{\mathrm{TILE}}\cdot H_{\mathrm{TILE}}=A_{\mathrm{TILE}}\cdot G_{\theta }\left(X\right)$$

Finally, the ultimate WSI representation $h_{\mathrm{WSI}}$ is fed into the classifier for obtaining the grade prediction, leveraging the region representations $H_{\mathrm{REG}}$ and the attention scores $A_{\mathrm{REG}}$ obtained from those same representations:(6)$$\hat{y}=\Psi_{\rho }\left(h_{\mathrm{WSI}}\right)=\Psi_{\rho }\left(A_{\mathrm{REG}}\cdot H_{\mathrm{REG}}\right)$$

## 5. Experiments

Within the scope of our experimental investigation, we systematically assessed and compared the impact of diverse magnification levels and their combinations, namely MONO, DI, and TRI-scale models. We further investigated the impact of several weakly supervised aggregation techniques in the performance of our deep learning model. These aggregation techniques vary in their ability to distill valuable diagnostic insights from the data, namely, mean, max, and attention-based. We put special emphasis on the comparison between a standard AbMIL architecture and the nested model proposed in NMGrad. Finally, we compared our solution to current state-of-the-art methods. The code is available at https://github.com/Biomedical-Data-Analysis-Laboratory/GradeMIL (accessed on 3 September 2024).

We list the details of the design choice of the models during training. VGG16 was used as the CNN backbone, with ImageNet pre-trained weights [48]. In our preliminary experiments, we explored various architectures and found that VGG16 exhibited favorable performance across multiple tasks on NMIBC WSIs. Stochastic gradient descent (SGD) was set as the optimizer, with a learning rate of 1 × 10^−4^, a batch size of 128, and a total of 200 epochs, with 30 epochs for early stopping, based on the AUC score on the validation set. A total number of 5000 tiles per WSIs were pre-emptively randomly sampled, to be further sub-sampled during training. Focal Tversky loss (FTL) was employed [49]. The Tversky index (TI) leverages false predictions, emphasizing on recall, in case of large class imbalance, tuning parameters α and β. TI is defined as
(7)$${\mathrm{TI}}_{c}\left(\hat{y},y\right)=\frac{1+\sum_{i=1}^{N}{\hat{y}}_{i,c}y_{i,c}+\epsilon }{1+\sum_{i=1}^{N}{\hat{y}}_{i,c}y_{i,c}+\alpha \sum_{i=1}^{N}{\hat{y}}_{i,\hat{c}}y_{i,c}+\beta \sum_{i=1}^{N}{\hat{y}}_{i,c}y_{i,\hat{c}}+\epsilon }$$
where ${\hat{y}}_{i,\hat{c}}=1-{\hat{y}}_{i,c}$ and $y_{i,\hat{c}}=1-y_{i,c}$ are the probability that sample *i* is not of class c∈C; ϵ is used for numerical stability, preventing zero division operations. FTL employs another parameter γ for leveraging training examples hardship:(8)$${\mathrm{FTL}}_{c}\left(\hat{y},y\right)=\sum {\left(1-{\mathrm{TI}}_{c}\left(\hat{y},y\right)\right)}^{1/\gamma }$$

## 6. Results and Discussion

The utilization of our proposed pipeline NMGrad using TRI-scale input emerged as a standout performer, as shown in Table 2. TRI-scale models showcased the ability to capture relevant patterns across different magnifications, substantially enhancing the overall grading accuracy. Comparing the effects of scales on the model performance shows a larger gap in performance between MONO and TRI models than structuring the processing of data using NMGrad or a standard AbMIL architecture. Furthermore, our exploration of aggregation techniques extended to mean and max aggregation methods, which did not include in-built attention mechanisms and yielded less promising outcomes. The absence of attention mechanisms rendered these techniques less effective in capturing nuanced features, underscoring the significance of attention mechanisms.

NMIA architecture embedded in NMGrad, which employs attention mechanisms for both tile and region aggregation, marked a substantial leap in performance in comparison to mean and max aggregation. This strategy provided the strengths of attention-based aggregation and ROI localization via a nested architecture. The incorporation of attention mechanisms allowed the model to pinpoint and emphasize critical visual cues within WSIs related to urothelial cell differentiation, ultimately resulting in a notable enhancement in predictive accuracy for bladder cancer grading. Finally, in a direct comparison to the previous best-performing model proposed by Wetteland [35], we implemented weakly supervised learning in a naive manner, where all patches were assigned a weak label. Predictions were made at the patch level, and the determination of WSI-level prediction relied somewhat arbitrarily on the summation of patch predictions, neglecting consideration of localized regions. Using our proposed solution NMGrad_TRI_, the solution aligned more closely with clinical expectations, such as the presence of one or more regions indicative of HG if the WSI was classified as HG, rather than scattered patches. Furthermore, the capacity of NMGrad_TRI_ to learn attention scores offered interpretability, as opposed to relying on ad hoc rules for post-processing patches into a final prediction. Ultimately, we obtained a slightly better F1 score, with a trade-off in accuracy. We also adhered to the works of Jansen [37] and Zhang [40] for comparing the performance of state-of-the-art grading algorithms, although the results corresponding to their in-house cohorts were different from ours. The development of our deep learning model NMGrad_TRI_ for predicting the grading of bladder cancer represents a significant advancement in the realm of accurate grading of bladder tumors.

In order to enhance the fidelity of binary decisions, we opted to introduce an uncertainty spectrum, thereby introducing a third class. Given the output predictions of test set WSIs, we defined the uncertainty spectrum as $\left[\mu_{\hat{y}\left(y=LG\right)}+\sigma_{\hat{y}\left(y=LG\right)},\mu_{\hat{y}\left(y=HG\right)}-\sigma_{\hat{y}\left(y=HG\right)}\right]$. A plot illustrating the concept and WSI predictions is shown in Figure 4. It was observed that if we were to exclude predictions falling within the uncertainty spectrum then the overall F1 score increased to 0.89. This underscores the potential utility of skepticism regarding non-confident predictions for robust clinical interpretation, which needs to be considered when implementing an algorithm at the inference stage.

Due to the attention scores generated at the inference stage, we were able to visualize a heatmap, as shown in Figure 5. NMGrad_TRI_ demonstrated effectiveness in correctly assessing individual tiles and ROIs, despite being trained solely on patient-level weak labels. We consulted the generated heatmaps with experienced pathologists for qualitative analysis. The results on the annotated set of regions, Test_ANNO_, exhibited competence in discerning LG and HG regions, as shown in Table 3. Region attention scores were considered for direct comparison to region prediction scores utilizing the classifier $\Psi_{\rho }$, as the values for both scores were restrained between 0 and 1. These two scores were evaluated individually against the annotated areas. It was corroborated that higher attention scores were associated with HG areas in high-grade WSIs. However, the same did not apply for low-grade. For low-grade WSIs, we observed a wide range of possibilities, where high attention was spread across regions. Ideally, one would anticipate a direct correlation between HG and elevated attention and, conversely, a correlation between LG and reduced attention. However, this correlation was primarily observable in the positive class (HG), aligning with the inherent design of MIL, which is tailored for identifying positive instances. In contrast to attention, we observed a high degree of correspondence between the label of annotated ROIs and the output region predictions.

We further investigated the correlation between region attention scores and the output region predictions, as displayed in Figure 6. We observed a generalized reciprocity between low-grade having smaller attention scores and prediction outputs, and vice versa. Moreover, the high-grade range of values was more limited to a lower range compared to the low-grade. For instance, low-grade areas with predictions rounding zero showed the broadest range of values. This observation aligns with the earlier statement, wherein the positive class typically exhibited a more focused distribution of attention scores, predominantly linked with positive HG instances. In contrast, the negative class dispersed attention across various regions within the WSI despite all presenting similar LG features. In regards to misclassified WSIs, we noted that LG WSIs manifested both high attention and prediction scores, whereas HG slides displayed a broad range of values. When examining the regression lines of TP and FP, they exhibited similar trends, as did TN and FN, respectively. Essentially, wrongly predicted WSIs exhibited characteristics contrary to their assigned class.

To augment the evaluation process, we integrated correlation calculations with follow-up information, thereby ensuring a more thorough assessment of our model’s performance. We employed the Cramér’s V correlation coefficient $\varphi_{c}$ for calculating the intercorrelation between grading and the event of recurrence and progression for the test set [50]. We observed a lack of correlation between grading and recurrence for either the manual or automatic grading, aligning with our expectations. However, for progression, NMGrad exhibited a higher correlation than the uropathologist (0.32 vs. 0.26, respectively). In accordance with [51], a strong correlation between grading and progression was observed. Notably, these correlations suggest that NMGrad’s grade may hold greater predictive value for assessing the likelihood of progression in the context of NMIBC.

## 7. Conclusions

Accurate grading of NMIBC is paramount for patient risk stratification, but it has long suffered from inconsistencies and variations among pathologists. Furthermore, the pathological workload is increasing, as well as its expenses. In response to this challenge, we have introduced the NMGrad pipeline, a pioneering approach in bladder cancer grading using WSIs. NMGrad starts by using a tissue segmentation algorithm, finding areas of urothelium in the slides. Thereafter, it leverages a nested AbMIL model architecture to precisely identify diagnostically relevant regions within WSIs and collectively predict tumor grade. Moreover, through a multiscale CNN model, NMGrad processes urothelium tissue tiles at multiple magnification levels. We observed that in high-grade patients, attention scores pinpointed specific ROIs, while in low-grade patients, attention was more dispersed, deviating from the expected MIL pattern. Our clinical evaluations demonstrated that NMGrad consistently outperformed previous state-of-the-art methods, achieving a 0.94 AUC score. This achievement represents a significant advancement in the field of bladder cancer diagnosis, with the potential to improve patient outcomes, reduce economic burdens, and enhance the quality of care in the management of this challenging disease.

## Figures and Tables

**Figure 1 bioengineering-11-00909-f001:**
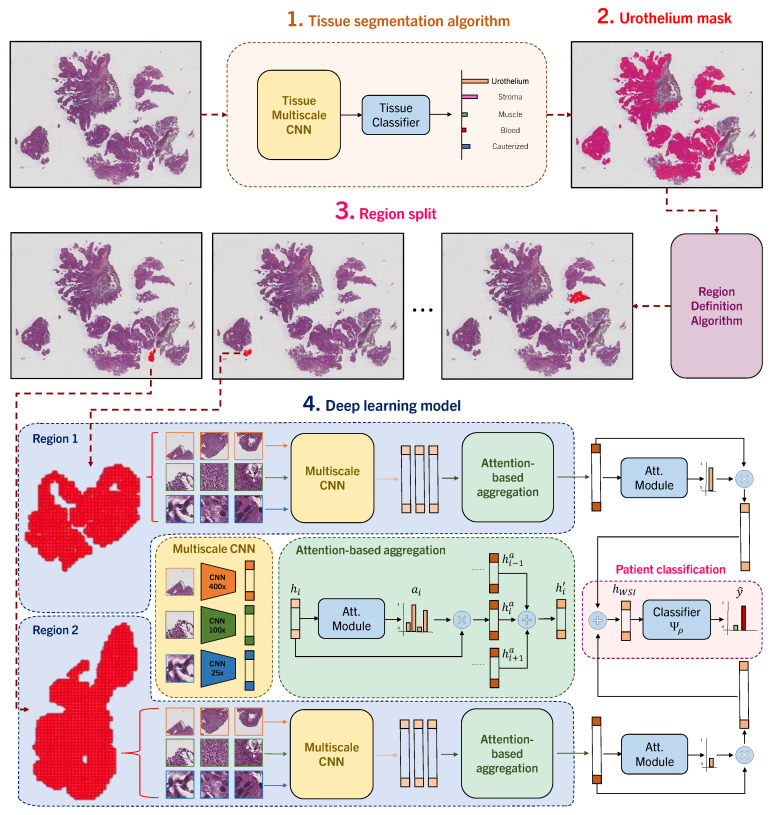
NMGrad pipeline. Initially, we apply a tissue segmentation algorithm for ROI extraction. Then, we pinpoint diagnostically significant urothelium areas within WSIs. Subsequently, we split the urothelium mask into regions, based on proximity and size, and extract tile triplets. In a hierarchical fashion, we further transform these triplets within their corresponding regions into region feature embeddings, using an attention-based aggregation method. All the region representations are then consolidated into a comprehensive WSI-level representation through a weight-independent attention module. Finally, this WSI feature embedding is input into the WHO04 grading classifier, in order to produce accurate WSI grade predictions.

**Figure 2 bioengineering-11-00909-f002:**
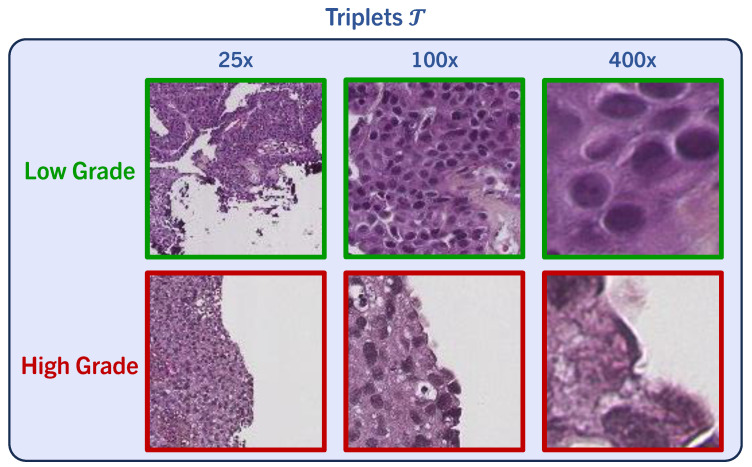
We obtain sets comprised of three tiles at different magnification levels, named triplets T, enabling detailed examination. Tile triplets demonstrate regions associated with low- and high-grade features.

**Figure 3 bioengineering-11-00909-f003:**
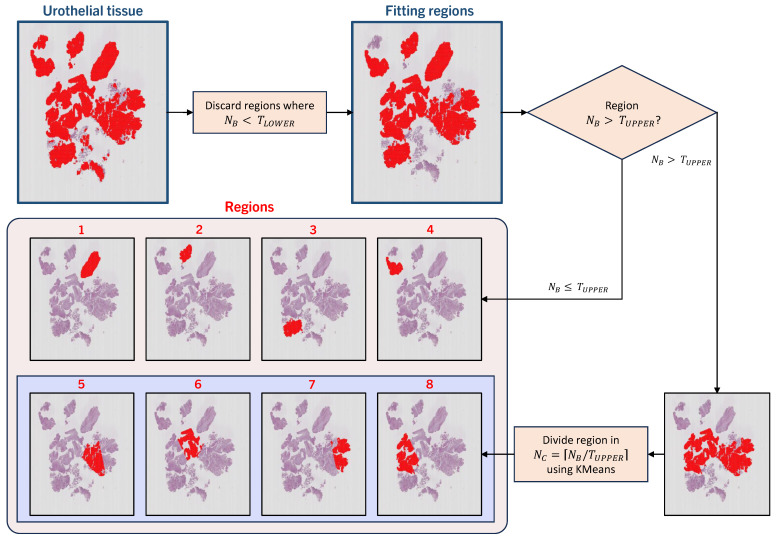
Region definition. Urothelial tissue within a WSI is eligible for tile extraction. Blobs of tiles are formed, and blobs smaller than a threshold $T_{\mathrm{LOWER}}$ are discarded. From the remaining blobs, any smaller than $T_{\mathrm{UPPER}}$ are kept and defined as a region. For blobs bigger than $T_{\mathrm{UPPER}}$, the blob is subdivided into smaller pieces, using the location of the individual tiles within and KMeans clustering. The obtained clusters are designated as regions.

**Figure 4 bioengineering-11-00909-f004:**
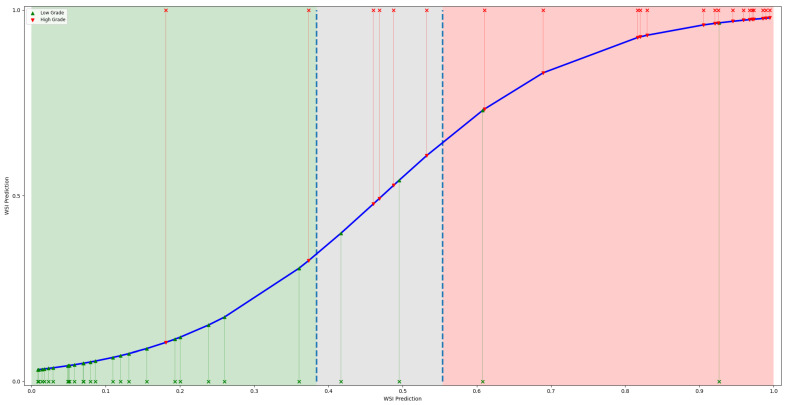
Plot displaying the WSI predictions of the test set, with green shading representing the LG confidence interval, red for HG, and gray denoting the uncertainty interval. Additionally, a blue line depicts the regression line fitting the predictions.

**Figure 5 bioengineering-11-00909-f005:**
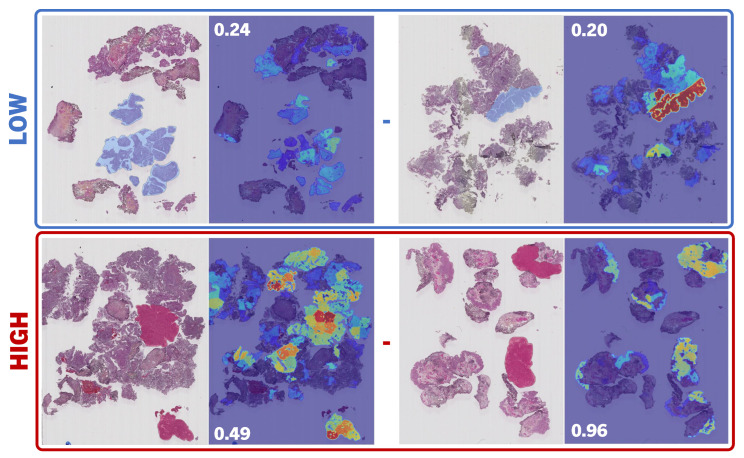
Region-level attention score heatmaps. Example regions of annotations of low- and high-grade ROIs annotated by a uropathologist are compared to the output attention provided by the proposed model NMGrad, left-to-right, respectively. The choice of annotated ROIs corresponded to the highest attention scores; red and blue correspond to low and high attention correspondingly. We have included the WSI-level prediction score for reference.

**Figure 6 bioengineering-11-00909-f006:**
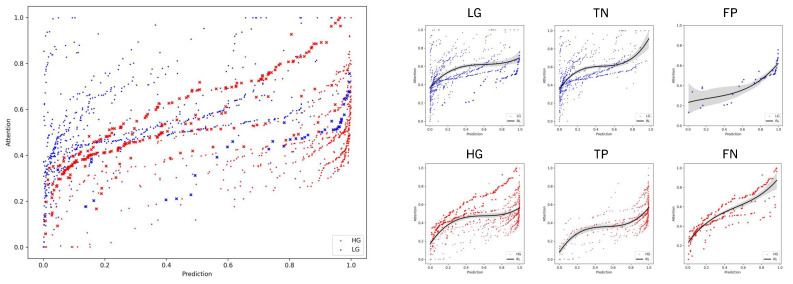
Correlation between region attention scores and output prediction of region embeddings on the test set of WSIs. Regions from accurately predicted WSIs (TN, TP) are denoted by squares, while those from incorrectly predicted WSIs (FN, FP) are marked with crosses. A discernible pattern emerges, where low attention scores align with diminished predictions and, conversely, higher attention scores correlate with elevated predictions. Additionally, an observable trend indicates that wrong predictions tend to manifest on the opposite end of the spectrum, with low-grade instances concentrating high attention and prediction scores, and vice versa. The trend is represented with a polynomial regression line (RL).

**Table 1 bioengineering-11-00909-t001:** Overview of the distribution of WSIs within each set, in terms of the WHO04 grading system ^1^.

Subset	Low-Grade	High-Grade
Train	124 (0)	96 (0)
Validation	17 (0)	13 (0)
Test	28 (7)	22 (7)

^1^ The number between parenthesis corresponds to slides containing some annotations giving region-based labels.

**Table 2 bioengineering-11-00909-t002:** Test performance for various aggregation techniques in weakly supervised learning. We provide the average of five runs, with the standard deviation shown in parentheses. The table presents the different approaches employed in the field, including aggregation techniques that involve considering spatial separation of the instances. We also explore the use of multiple magnification levels, considering 400× as the foundation for all magnification analysis. We also show the results from other bladder cancer grading works, although in other datasets.

Model	Accuracy	Precision	Recall	F1 Score	κ	AUC
**AbMIL_MONO_**	0.68 (0.07)	0.71 (0.07)	0.68 (0.06)	0.67 (0.07)	0.36 (0.12)	0.81 (0.07)
**AbMIL_DI_**	0.79 (0.09)	0.80 (0.10)	0.78 (0.09)	0.78 (0.09)	0.57 (0.18)	0.85 (0.13)
**AbMIL_TRI_**	0.82 (0.07)	0.82 (0.07)	0.82 (0.07)	0.82 (0.07)	0.64 (0.14)	0.91 (0.04)
**MEAN_TRI_**	0.81 (0.03)	0.83 (0.03)	0.80 (0.03)	0.80 (0.03)	0.61 (0.05)	0.92 (0.03)
**MAX_TRI_**	0.80 (0.06)	0.80 (0.06)	0.78 (0.06)	0.79 (0.07)	0.58 (0.13)	0.85 (0.06)
**NMGrad_MONO_**	0.68 (0.09)	0.71 (0.08)	0.69 (0.08)	0.68 (0.09)	0.37 (0.16)	0.80 (0.06)
**NMGrad_DI_**	0.83 (0.03)	0.85 (0.03)	0.82 (0.03)	0.82 (0.03)	0.65 (0.06)	0.91 (0.04)
**NMGrad_TRI_**	**0.86 (0.03)**	**0.87 (0.02)**	**0.85 (0.04)**	**0.85 (0.03)**	**0.71 (0.06)**	**0.94 (0.01)**
**Wetteland [35]**	0.90 (-)	0.87 (-)	0.80 (-)	0.83 (-)	-	-
**Jansen [37]**	0.74 (-)	-	0.71 (-)	-	0.48 (0.14)	-
**Zhang [40]**	0.95 (-)	-	-	-	-	0.95 (-)

**Table 3 bioengineering-11-00909-t003:** Region-level prediction and attention correspondence on annotated areas, using NMGrad_TRI_. We individually compared the degree of consensus of annotations with both the highest attributed attention within the WSI and the output region prediction of the classifier $\Psi_{\rho }$.

Output	Accuracy	Precision	Recall	F1 Score
Attention	0.76	0.81	0.69	0.75
Prediction	0.89	0.83	0.91	0.87

## Data Availability

The raw data underlying this article were generated at Stavanger University Hospital. Derived data supporting the findings may be available from the corresponding author upon request.

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
