# Peer review of "NMGrad: Advancing Histopathological Bladder Cancer Grading with Weakly Supervised Deep Learning"

_bioengineering, 2024, doi:10.3390/bioengineering11090909_

Round 1

Reviewer 1 Report

Comments and Suggestions for Authors

This is a very well written manuscript. Authors proposed a weakly supervised deep learning algorithm to grade bladder cancer into low and high grade groups.  

The background and related works gave enough information and detail to cover the scope of the project while providing insight into the current state of computational pathology (what they call CPATH) in bladder cancer diagnosis. 

I particularly liked how they emphasized the role of attention scores and how the utilization of such scores led to enhanced interpretability and deduction of specific tissue region(s) that lead to the diagnosis of low/high grade.

They also revisited the attention score in the discussion, and looked at how their scores were reflected in the incorrectly predicted whole slide images (WSI). 

I have 2 comments I believe would improve the quality of this manuscript.

-          Table 2. I would like to have the raw number for at least the NMGradTri. What are the False Negative, FP, TN and TP values  ? or even better could the authors give the matrix of confusion. It would be interesting to know if misclassification rate is the same for low and high grade lesions

-          Even though the result son the independent test are quite impressive, we are all aware (including the authors) that the main issue with deep learning algorithms is their generalization to external data set. As stated in line 161-12 “all WSI originate form the same laboratory resulting in relatively consistent staining color across the dataset”, all slides were stained in the same lab but also scanned with the same scanner. I am surprized that the authors do not even discuss this issue in their discussion and conclusion.  It would be great if the authors could rescan the test slides on another scanner and compare the performance of their algorithm. But at least It would make this manuscript better if authors could add few sentences on this issue. Again it is hard for me to imagine that the authors, based on the quality of this work, omitted to discuss this problem.  What is the next step? Are they planning to test their algorithm to a set of bladder cancer form a different institution stained in a different lab.  Any use of their algorithm requires first to prove that it can be validated to an external dataset

One small edit on line 25 on the first page - 'not only' should be removed.

) management of NMIBC, since treatment strategies not only rely on the presence of 

Reviewer 2 Report

Comments and Suggestions for Authors

The authors present a technique for grading bladder cancer on histological slides using nested multiple instance learning incorporating attention scores. Although the use of a single institution improves the consistency of the data, eventual validation in external datasets may be needed to evaluate generalizability. A study strength is that code is provided. In addition, the performance is compared with other studies in the literature.

The non-published material is not in English.

Abstract, Line 4: “Moreover, absence of annotations in medical imaging difficults training deep learning models.” Do you mean, “Moreover, the absence of annotations in medical imaging creates difficulty in training deep learning models.”

Data Material, Line 163: “All WSIs were graded by an expert uropathologist in accordance with the WHO04 classification system, as either LG or HG, thus providing slide-level diagnostic information. However, the dataset lacks region-based annotations pinpointing the precise areas of LG or HG regions within the WSI. Consequently, the dataset is considered weakly labeled. For WSIs labeled as LG, at least one LG region is expected, with the possibility of presenting non-cancerous tissue in other regions. As for HG slides, at least one region should display HG tissue, while other regions may exhibit a LG appearance or non-cancerous tissue. Given the absence of alternative gold standards, we are compelled to continue utilizing a grading assessment that may have limitations for training and evaluating our algorithms.” Has a similar labeling scheme (slide-level) been used for other studies applying deep learning for histological images?

Data Material, Line 173: “The split employed ensures that each subset maintains the same proportional representation of diagnostic outcomes. This stratification encompassed factors such as WHO04 grading, cancer stage, recurrence, and disease progression to best mirror the diversity of the original data material. The distribution of LG and HG WSIs within each dataset is detailed in Table 1 for reference.” It may be helpful to include these other factors in table 1 (recurrent, disease progression). 

Table 1: “The number between parenthesis corresponds to slides containing some annotations giving region-based labels.” Please clarify what you mean by this.

Methods, Region Definition: How are Tupper and Tlower determined?

Results & Discussion: Other works are compared, but they use a different dataset. Could there be some future utility in establishing a benchmark dataset for this task?

Results & Discussion, Line 320: “It was observed that, if we were to exclude predictions falling within the uncertainty spectrum, the overall F1 score increased to 0.89.” Would it be possible to provide other performance metrics when the uncertainty spectrum is introduced?

Figure 5: “The choice of annotated ROIs correspond to highest attention scores, for red and blue correspond to low and high attention correspondingly.” To clarify, in the top row, the pathologist annotated low grade tissue in blue and in the bottom row, the pathologist annotated high grade tissue in red? I presume the attention maps are the ones with blue background: is high attention on them red (rather than blue)?

Could these heatmaps be used to aid pathologists when they are evaluating slides? With further development, I would imagine that they could potentially aid in diagnostic interpretation.

Figure 6: It is a bit hard to appreciate squares on these images, as they look more like dots due to the resolution.

Results & Discussion, Line 344: “Moreover, the high-grade range of values is more limited to a lower range compared to the low-grade. For instance, low-grade areas with predictions rounding zero show the broadest range of values.” By range of values, I presume you mean attention scores.

Results & Discussion: The article could benefit from a discussion of strengths and weaknesses, as well as commenting upon future directions. A study future direction could be to extend this analysis to external datasets.

General: Could NMGrad be useful for assessing histological grading of other forms of cancer in the future?

Comments on the Quality of English Language

This article could benefit from some minor editing of the English/grammar.

Round 2

Reviewer 2 Report

Comments and Suggestions for Authors

The authors have addressed my concerns.

One minor point: in Figure 5, does red correspond to low attention and blue correspond to high attention, or is it vice-versa?